# Multi-Component Intervention to Promote Physical Activity in Japanese Office Workers: A Single-Arm Feasibility Study

**DOI:** 10.3390/ijerph192416859

**Published:** 2022-12-15

**Authors:** Jihoon Kim, Ryoko Mizushima, Kotaro Nishida, Masahiro Morimoto, Yoshio Nakata

**Affiliations:** 1Graduate School of Comprehensive Human Sciences, University of Tsukuba, 1-1-1 Tennodai, Tsukuba 305-8574, Japan; 2Department of Sports Research, Japan Institute of Sports Sciences, 3-15-1 Nishigaoka, Kita-ku, Tokyo 107-0061, Japan; 3Risk Management Department 4th, MS&AD InterRisk Research & Consulting, Inc., WATERRAS ANNEX (10F & 11F), 2-105, Kanda Awajicho, Chiyoda-ku, Tokyo 101-0063, Japan; 4Faculty of Health and Sport Sciences, University of Tsukuba, 1-1-1 Tennodai, Tsukuba 305-8574, Japan

**Keywords:** exercise, workplace, health promotion, behavioral research, occupational health

## Abstract

This study investigated the feasibility of a multi-component intervention to promote physical activity (PA) among Japanese office workers. It was an 8-week single-arm trial conducted in Japan in 2021, in which 76 employees aged 20 or older, from an insurance company, participated. They received a multi-component PA intervention that comprised individual (lecture, print material, goal setting, and feedback), socio-cultural (team building and supportive atmosphere), physical (poster), and organizational (encouraging message from an executive) strategies. The primary outcome was change in objectively measured moderate-to-vigorous PA (MVPA). A paired t-test was used to compare the changes between weeks 0 and 8. We also conducted a subdomain analysis of PA divided into four domains (working, non-working, commuting working, and remote working). Excluding 26 participants who could not complete valid assessments, the MVPA among participants (*n* = 50, age 49.6 ± 9.7) significantly increased by +7.3 min/day [95% confidence interval (CI) 0.8 to 13.8]. We also identified significant changes in MVPA by +10.0 min/day [95% CI, 3.7 to 16.3] in working days (*n* = 40), and by +7.1 min/day [95% CI, 0.4 to 13.7] in remote working days (*n* = 34). We demonstrated that multi-component PA interventions might improve MVPA among Japanese office workers.

## 1. Introduction

Physical inactivity is a major public health problem and the leading cause of mortality and non-communicable diseases such as cardiovascular diseases, diabetes, and cancer. It has also been associated with poor mental health and well-being [1,2]. The global economic burden induced by physical inactivity is estimated at 53.8 billion US dollars, which has an effect on public health systems and individual financial status [3]. To tackle the pandemic of physical inactivity, the World Health Organization (WHO) and many developed countries have announced guidelines to promote physical activity (PA) [4,5,6,7]. Despite these efforts, the prevalence of insufficient PA (the global recommendation on PA for health is at least 150 min of moderate-intensity PA per week, at least 75 min of vigorous-intensity PA per week, or an equivalent combination) in adults was 27.5% worldwide and 30–39.9% in Japan [8]. Factors that contribute to physical inactivity in modern adults are the increased use of passive modes of transportation such as motor vehicles, being active during leisure time, and an increased sedentary lifestyle both at home and work [1,9]. In the past few decades, there has been a notable shift from manual tasks to sedentary tasks at work and sedentary jobs have increased [9]. As adults spend half of their waking time in the workplace [10], these changes in occupational settings could be considered the main cause of physical inactivity for adults. In particular, office workers have a high health risk because they spend much of their daily life with low PA and high sedentary time (ST) compared to other occupational groups [11,12]. Previous studies have also reported that workers with low PA and high ST may have poor work-related outcomes [13,14,15].

The workplace has recently been recognized as an important public setting in im-proving PA [16]. The WHO has recommended a workplace-based approach to improve PA by considering workers’ daily lives [16]. Many studies have identified the effectiveness of workplace-based interventions in improving PA [17,18,19,20]. The components of the intervention programs are lectures, self-monitoring, exercise programs, counseling, goal setting, print material, pedometers, incentives, group support, and treadmill workstations [17,18,19,20,21]. Moreover, compared to single-component or environmental interventions, multi-component interventions have been reported to be more effective [17,18]. Despite many people perceiving the benefits of PA intervention in the workplace, there are barriers such as socio-ecological reasons and differences in perspective between employers and employees [22,23,24]. Therefore, it is crucial to develop and conduct intervention programs that consider socio-ecological reasons among office workers’ and stakeholders’ opinions in the workplace. Meanwhile, the COVID-19 pandemic has caused people to be even more physically inactive [25]. It has affected people’s health and the occupational environment (i.e., converting commuting work to remote work) [26,27]. In Japan, many companies have recently adopted remote work to promote social distancing, which may further increase the physical inactivity of workers. In particular, office workers remain vulnerable to social and natural environmental changes because their low PA and high ST daily lifestyles may worsen with changes from commuting work to remote work. Further, conventional strategies in offices for improving PA will not be effective or realistic due to these changes in daily life caused by the pandemic. Thus, an intervention program to improve PA in office workers should correspond to the new normal of life with COVID-19.

In a previous study, we developed, through a focus group interview, a comprehensive and multi-component intervention to promote PA among Japanese office workers. Our main goals were to investigate (1) workers’ perception of the importance of PA, (2) the current status of PA promotion programs available in the workplace, and (3) the facilitators of, needs for, feasibilities of, and barriers to PA promotion interventions [28]. We proposed the following intervention programs: individual (information delivery), socio-cultural environment (team building, supportive atmosphere), physical environment (standing desk, poster), and organizational (incentive, encouraging messages from an executive, workplace policy) strategies. However, there was no evidence that our proposal could promote PA among office workers. Thus, in the present study we evaluated the feasibility of the proposed program among Japanese office workers.

## 2. Materials and Methods

### 2.1. Design and Participants

This 8-week, single-arm feasibility study was conducted in Japan from January to March 2021 during the Japanese government’s second state of emergency. The participants belonged to an insurance company located in Tokyo or branches in Osaka, Nagoya, Fukuoka, and Sapporo, and were recruited through an internal newsletter. Conventionally, their work was office based. However, due to the changes caused by the COVID-19 pandemic, they had switched to flexible work, either remote or office-based, based on their self-decision or recommendations from their departments. The eligibility criteria for participation in the study was being an office worker between ages 20 and 65 regardless of sex. Workers who planned to permanently or temporarily retire during the intervention period, who could not use smartphones, and those who had gait disturbance due to injury or other reasons were excluded from the sample. Written informed consent was obtained from all the participants and approval for the study was obtained from the University of Tsukuba Faculty of Health and Sport Sciences Ethics Committee (approval number Tai 020-149).

### 2.2. Intervention Program

The participants received an 8-week multi-component intervention program remotely to improve their PA. It was based on our previous study, the proposal of a comprehensive and multi-component PA intervention [28], and was composed through deliberation with the participating company. First, we proposed various intervention programs that could be adopted at their worksite by the participating company’s research staff (K.N. and M.M.). Second, company representatives discussed the intervention program with the wellness committee of the firm and chose programs that could be introduced without much financial or occupational environmental charge. Finally, after further deliberation with the company, we offered the participants the following multi-component PA intervention program (Figure 1). Each PA program was delivered remotely, and the participating company’s research staff (K.N.) was supported on-site.

#### 2.2.1. Individual Strategy

We provided participants with lectures, print materials, goal setting, and feedback as individual strategies. In week 1, they received a 30 min motivational lecture given by a researcher (Y.N.) and print material (PDF file of lecture contents) through an online. The contents of the lecture comprised the definition of PA, evidence of “+10 min of Physical Activity per Day” [29], recommendation of appropriate exercise intensity, the difference between physical inactivity and sedentary behavior [30], and the importance of breaking sedentary behavior in daily life [31]. Additionally, the participants received tips on setting goals to improve PA.

We also provided participants with feedback consisting of a brief weekly message and a summary report containing the current state of their objectively measured PA and ST. Brief weekly messages written by a researcher (Y.N.) were sent through internal e-mails. The messages contained more detailed action plans to improve PA and ST, and they were delivered to the participants every Monday a total of seven times from week 2 to week 8. The participants also received a 7-day PA and ST summary report of baseline results in week 2 to inform the current status of their PA and ST. It contained their daily steps, activity score (metabolic equivalents × hours per day), and number of sit-to-stand measurements calculated using a thigh-worn accelerometer (PAL Technologies, Glasgow, UK). This thigh-worn accelerometer was offered during the baseline assessment; and it was not used for outcome measurement, but rather to give participants feedback on their current PA and ST.

#### 2.2.2. Socio-Cultural Environment Strategy

We provided the participants with a supportive atmosphere and team building as part of their socio-cultural environmental strategies. After starting the intervention, the participating company’s research staff (K.N.) informed the participants of the intervention program contents, schedules, and prompts to create a supportive atmosphere through internal e-mail. This prompt was delivered every Monday from week 2 to week 8, combined with a short message by a researcher (Y.N.). The participants participated in individual (week 4) and team-based (week 6) step-count competitions for team building using smartphone applications provided by the company. The number of participants per team was approximately four, and the teams were decided by research staff (K.N.) based on their department. The results of the individual and team-based competitions were announced through an internal newsletter in weeks 5 and 7, respectively.

#### 2.2.3. Physical Environment Strategy

We posted PA prompt posters in the office and sent them as PDF files to those who worked remotely through internal e-mails four times during the intervention period. The contents of the PA prompt poster were (1) the health risks associated with prolonged sitting time, (2) the health benefits of replacing sitting with standing, (3) the health benefits of walking, and (4) the importance of goal setting for improving physical activity in weeks 1, 3, 5, and 7, respectively.

#### 2.2.4. Organizational Strategy

The organizational strategy involved an executive announcing an encouraging message to the participants; he participated in the intervention program as well. Encouraging messages and recommendations by executive were delivered through the lecture in week 1 and the internal newsletter in weeks 5 and 7.

### 2.3. Measurements

#### 2.3.1. Baseline Characteristics

The participants reported their basic characteristics, including sex, age, height, weight, current smoking status, education level (four-year college graduate, college graduate, or high school graduate or less), continuous years of service, living arrangement (living alone or living with one or more other people), and marital status (married or not). Body mass index (BMI) was calculated as a participant’s weight (kg) divided by their squared height (m^2^). Each questionnaire item was optional, so participants were not forced to answer if they did not want to. All the self-administered questionnaires were conducted via an online survey.

#### 2.3.2. Primary and Secondary Outcomes

The primary outcome was the change in moderate-to-vigorous physical activity (MVPA) over 8 weeks. The secondary outcomes were changes in light PA (LPA), moderate PA (MPA), vigorous PA (VPA), steps, and ST. The participants wore a triaxial accelerometer (Active style Pro HJA-350IT; Omron Healthcare, Kyoto, Japan) for seven consecutive days. This device is a small accelerometer worn on the waist, which counts the participants’ daily steps and PA intensity (expressed as metabolic equivalents) through a validated algorithm [32,33]. We asked the participants to detach the device while sleeping, bathing, swimming, or participating in contact sports for device status and safety reasons. Referencing previous studies, we defined valid data as wearing time of at least 10 h per day and wearing days for at least 3 days during the wearing period [34,35]. Finally, the participants were instructed to keep a daily diary to note their wake-up time, bedtime, working time, break time, and type of work (remote or commuting). The outcome measures were assessed at weeks 0 and 8.

#### 2.3.3. Post-Intervention Survey

We conducted an additional post-intervention survey to obtain the participants’ evaluations of the intervention program. The participants were asked to grade the program on a 5-point satisfaction scale (very satisfied, satisfied, neither, unsatisfied, and very unsatisfied). We also asked them: “What component of the intervention do you think was most effective in improving physical activity?”

### 2.4. Statistical Analysis

We presented the participants’ baseline characteristics as means (with standard deviations) for continuous variables or as frequencies and percentages for categorical variables. A paired t-test was used to compare the changes in objectively measured MVPA, LPA, MPA, VPA, steps, and ST between weeks 0 and 8. We also conducted sub-domain analyses divided into working days, non-working days (i.e., weekends or holidays), commuting working days, and remote working days to identify domain-specific differences in PA. Primary and secondary outcomes were presented as means and 95% confidence intervals (CI). All statistical analyses were conducted using SPSS version 28.0 (IBM Corp., Armonk, NY, USA), with the level of statistical significance set at 5%.

## 3. Results

A participant flowchart is shown in Figure 2. We initially approached 274 office workers affiliated with an insurance company for recruitment through an internal informational announcement. Of them, 76 agreed to participate and provided written informed consent. For technical reasons, we could not hand accelerometers to participants from offices other than Tokyo. Therefore, we excluded these participants from the analysis (Osaka 5, Nagoya 2, Fukuoka 2, and Sapporo 2). In addition, a participant in the Tokyo office did not report basic characteristics. It left 64 participants, who completed the baseline assessment, the 8-week multi-component intervention program, and the 8-week follow-up assessments. The baseline characteristics of the study participants are shown in Table 1. The mean age of the participants was 49.3 (9.3) years, 84% of the participants were men, and most were four-year college graduates (95%).

Table 2 presents the PA outcomes of the intervention program. We identified the significant increase in MVPA (+7.3 min/day [95% CI, 0.8 to 13.8]), MPA (+6.6 min/day [95% CI, 0.3 to 13.0]), and steps (+873 steps/day [95% CI, 169 to 1576]). We did not identify any significant changes in the other outcome measures.

Table 3 presents the results of the sub-domain analyses, which divided a week (seven consecutive days) into working and non-working days. Significant increases were found in MVPA (+10 min/day [95% CI, 3.7 to 16.3]), MPA (+9 min/day [95% CI, 2.8 to 15.1]), VPA (+1 min/day [95% CI, 0.01 to 2.1]), and steps (+1172 steps/day [95% CI, 365 to 1979]) on working days. Meanwhile, only steps significantly increased (+1310 steps/day [95% CI, 63 to 2557]) on nonworking days. Other outcome measures did not significantly change on either working or non-working days (*p* > 0.05).

Table 4 presents the results of the subdomain analyses, which divided working days into commuting and remote working days. Significant increases were found in MVPA (+7.3 min/day [95% CI, 0.5 to 14.2]), MPA (+7.5 min/day [95% CI, 0.9 to 14.1]), and steps (+822 steps/day [95% CI, 17 to 1627]) on remote working day. Other outcome measures did not significantly change on either commuting or remote working days (*p* > 0.05).

Finally, the post-intervention survey data is presented in Figure 3. Among the 64 participants who completed the 8-week follow-up assessment, 54 answered the post-intervention survey. On a 5-point satisfaction scale used to rate the intervention program, “very satisfied,” “satisfied,” and “neither” received 31, 13, and 10 responses, respectively. None of the participants answered “unsatisfied” and “very unsatisfied.” The proportion of participants who answered ‘yes’ to ‘Do you think that each component of the intervention was effective in improving PA?’ was 57% for goal setting and team building, and 78% for encouraging messages from an executive.

## 4. Discussion

The present single-arm study evaluated the feasibility of a multi-component intervention to improve PA in Japanese office workers. The 8-week multi-component PA intervention consisted of individual (lecture, print material, goal setting, and feedback), socio-cultural environment (team building and supportive atmosphere), physical environment (poster), and organizational (encouraging message from an executive) strategies using non-face-to-face remote systems. This study resulted in an improvement in MVPA (+7.3 min/day, our primary outcome), MPA (+6.6 min/day), and steps (+873 steps/day) over 8 weeks. In particular, we found significant improvement in MVPA, not in LPA (+1.7 min/day), contrary to previous studies that conducted workplace multi-component PA promotion and reported improvement in LPA [36,37,38].

We assumed that this significant change in MVPA might have occurred because of the following aspects. First, the participants had high health literacy as their education level was high (four-year college graduates 95%) and they worked for an insurance company. A systematic review reported that high health literacy was positively associated with high PA levels [39]. Therefore, the participants might have responded more readily to the PA program than those of previous studies. Second, the PA intervention program was offered after carefully considering the current status and needs of workers of the insurance company through our previous interview study [28]. Therefore, it was probably very suitable in promoting PA among them. Finally, environmental changes in the COVID-19 infection status during the measurement period might have affected the results. In the present study, the timing of the baseline measurement (week 0) and the post-intervention measurement (week 8) corresponded to the first and last weeks of the Japanese government’s second state of emergency (from 8 January 2021 to 20 March 2021). According to a previous study [40], PA decreased immediately after the lockdown began but tended to recover as the end of lockdown approached. Thus, the environmental changes caused by COVID-19 may have influenced our results.

Another study reported that workers who take remote work are low in MVPA and LPA and high in ST compared to those who commuted to work [41]. The participants of our study had flexible working situations (remote or commuting) to cope with the COVID-19 pandemic. Therefore, we conducted a sub-domain analysis divided into working and non-working days, and commuting working and remote working days. In the subdomain analysis comparing working and non-working days, significant improvements were found in MVPA (+10 min/day), MPA (+9 min/day), VPA (+1 min/day), and steps (+1172 steps/day) on working days. In particular, approximately +10 min/day improvement in MVPA on working days is a very meaningful result as this amount of change has been reported to provide benefits for the risk of NCDs, mental health, musculoskeletal disease, and mortality [29,42]. Meanwhile, a significant improvement was found in steps of +1310 steps/day on nonworking days. The other outcomes were not significant, partly because of the high baseline measurements. For example, the baseline MVPA on non-working days was higher (63.7 min/day) than that on working days (45.8 min/day). This amount of MVPA meets the Japanese PA guideline, which recommends MVPA for 60 min every day, regardless of the type of PA [6].

In the sub-domain analysis comparing commuting and remote working days, significant changes were observed in MVPA (+7.1 min/day), MPA (+7.3 min/day), and steps (+826 steps/day) on remote working days. In contrast, no significant changes were observed on commuting working days. One reason why the significant changes were observed on remote working days was the considerable baseline difference of MVPA between commuting and remote working days (58.5 min/day vs. 34.1 min/day, *p* < 0.001). Although a higher level of MVPA means less room for improvement on commuting working days, there is scope for improvement on remote working days. Moreover, on remote working days, workers may have more leisure time than on commuting working days. Thus, it might have been easy for the participants to replace the sedentary leisure time with MVPA.

We also conducted a post-intervention survey to ascertain participants’ evaluations of the intervention program. According to the literatures, workers need various intervention programs for various socio-ecological reasons, including individual, environmental, and organizational aspects [22,23,24,28]. In this study, we provided a multi-component PA intervention considering the above-mentioned socio-ecological concerns, and more than half of the participants reported that each component of the intervention program was effective. This finding is in line with those of previous studies [21,43,44]. More than 70% of the participants reported that encouraging messages from an executive (organizational strategy), posters (physical environment strategy), a supportive atmosphere (socio-cultural environment strategy), and lectures (individual strategy) were effective. Therefore, the present programs are assumed to be appropriately composed in terms of the socio-ecological context of the participants.

The strengths of the present study are as follows. First, we identified the feasibility of a non-face-to-face PA intervention program in a “new normal” routine under COVID-19 restrictions [45]. Second, the present intervention program is cost-effective compared to previous studies, which provided incentives, sit-stand workstations, treadmill workstations, and wearable devices [18,19,20,21]. The lower financial and occupational burden of our program is more likely to be acceptable and feasible for most companies. Finally, we identified the different effects of remote and commuting working days on PA and demonstrated a significant improvement in MVPA on remote working days. To our knowledge, few intervention studies have analyzed changes in PA on remote and commuting working days, despite the dissemination of remote work. Therefore, this study is meaningful for contemporary workers promoting PA.

Nevertheless, the present study had several limitations. First, it was a single-arm feasibility trial without a control group. Therefore, it was impossible to distinguish between the effect of the treatment and that of any number of confounding variables [46]. Thus, a well-designed randomized controlled trial is necessary to evaluate the effectiveness of this intervention program. Second, as the present intervention was conducted in the short term, it is impossible to predict its long-term effectiveness and sustainability. Longer intervention and follow-up periods are necessary to address this issue. Third, we could not identify which component of the intervention program was effective; it may be dependent on the participants’ individual characteristics. Further qualitative studies could clarify this issue. Finally, the findings of the present study are difficult to generalize because all the participants were from the same insurance company, most (95%) were four-year college graduates, and the majority (83%) were men. Further studies in other firms are necessary to implement and disseminate the intervention program.

## 5. Conclusions

The present study suggests that a multi-component intervention program promotes PA among Japanese office workers. Moreover, our sub-domain analyses suggest that the effects on PA are different among on working, non-working, commuting, and remote working days. Further long-term studies and randomized control trials are necessary to clarify its efficacy.

## Figures and Tables

**Figure 1 ijerph-19-16859-f001:**
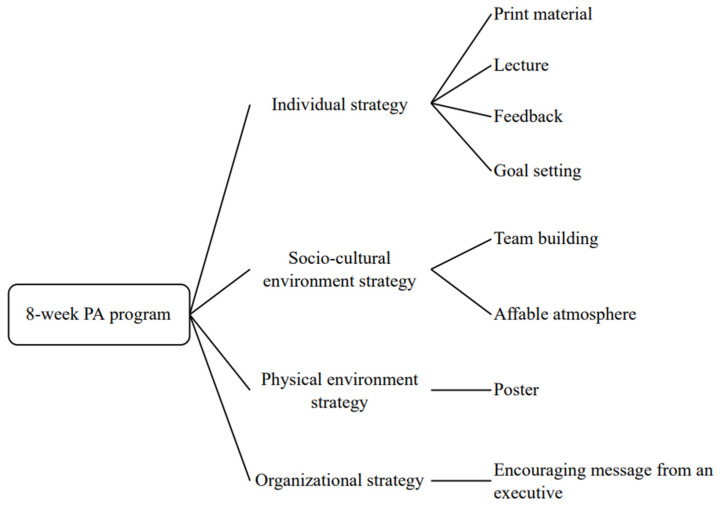
Comprehensive and multi-component PA program.

**Figure 2 ijerph-19-16859-f002:**
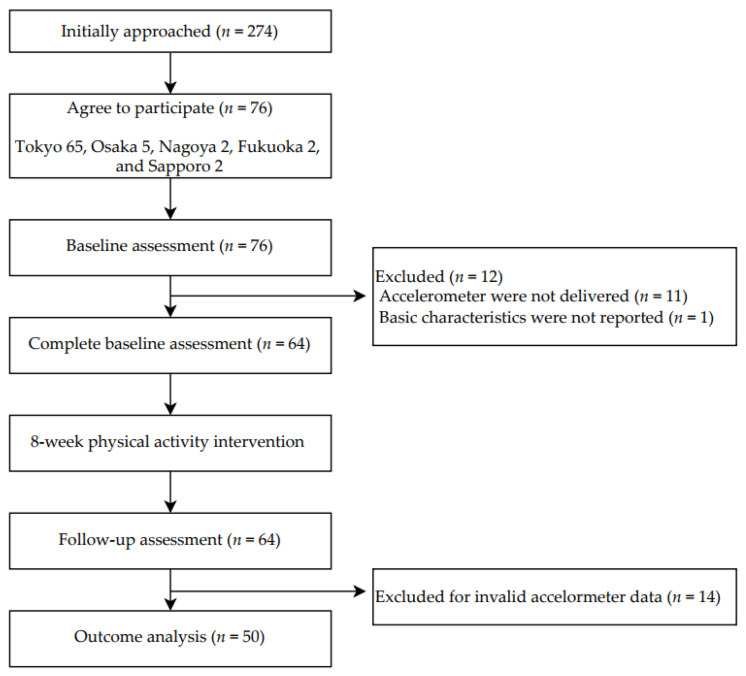
Participant flowchart.

**Figure 3 ijerph-19-16859-f003:**
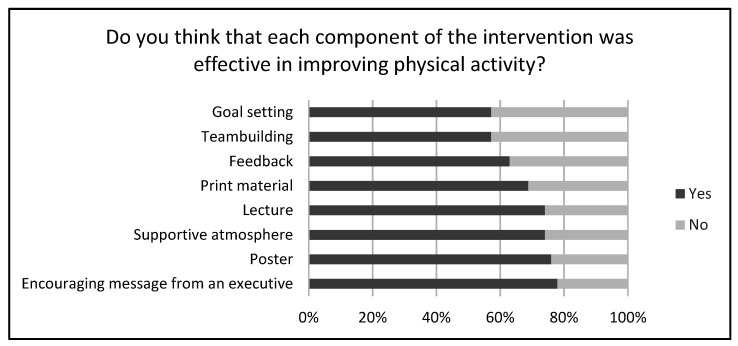
Post-intervention survey (*n* = 54).

**Table 1 ijerph-19-16859-t001:** Characteristics of participants at baseline.

Characteristic	Participants (*n* = 64)
Age, years	49.6 (9.3)
Women, *n* (%)	10 (16)
Height, cm	171.9 (7.4)
Weight, kg; *n* = 61 *	69.8 (12.9)
Body mass index, kg/m^2^; *n* = 61 *	23.3 (3.3)
Current smoker, *n* (%); *n* = 59 *	6 (10)
Four-year college graduate, *n* (%)	61 (95)
Continuous years of service, years; *n* = 59 *	10 (0, 37)
Living with one or other people, *n* (%); *n* = 59 *	55 (93)
Currently married, *n* (%); *n* = 59 *	52 (89)

Notes: Data are presented as mean (standard deviation) for age, height, and body mass index, median (range) for continuous years of service, and number (%) for categorical variables. * Missing data were mostly due to personal reasons (they did not want to answer).

**Table 2 ijerph-19-16859-t002:** Physical activity outcomes during the intervention program (*n* = 50).

	Week 0	Week 8	Change (95% CI)	*p*-Value
Characteristic				
Age, years	49.6 (9.8)			
Women, *n* (%)	10 (20)			
BMI, kg/m^2^; *n* = 47 *	22.8 (3.1)			
Physical activity outcomes				
MVPA, min/day	50.0 (27.1)	57.3 (29.8)	7.3 (0.8 to 13.8)	0.028
LPA, min/day	196.1 (64.6)	197.8 (67.9)	1.7 (−10.8 to 14.3)	0.786
MPA, min/day	48.9 (26.8)	55.5 (29.5)	6.6 (0.3 to 13.0)	0.042
VPA, min/day	1.1 (3.2)	1.8 (4.8)	0.7 (−0.1 to 1.4)	0.071
Steps, steps/day	6701 (2859)	7574 (3003)	873 (169 to 1576)	0.016
ST, min/day	573.4 (85.2)	566.6 (89.2)	−6.9 (−27.9 to 14.2)	0.515
Valid day	6.9 (1.7)	6.8 (1.4)	−0.1 (−0.7 to 0.4)	0.654

Notes: Data are presented as mean (standard deviation) for continuous variables or number (%) for categorical variables at weeks 0 and 8, and mean (95% CI) for change. Abbreviations: BMI, body mass index; CI, confidence interval; LPA, light physical activity; MPA, moderate-to-vigorous physical activity; ST, sedentary time; VPA, vigorous physical activity. * Missing data were mostly due to personal reasons (they did not want to answer).

**Table 3 ijerph-19-16859-t003:** Sub-domain analyses on working day and non-working day (*n* = 40).

	Week 0	Week 8	Change (95% CI)	*p*-Value
Characteristic				
Age, years	51.1 (8.9)			
Women, *n* (%)	8 (20)			
BMI, kg/m^2^; *n* = 38 *	22.5 (2.9)			
Physical activity outcomes on working day				
MVPA, min/day	45.8 (27.4)	55.7 (28.8)	10.0 (3.7 to 16.3)	0.003
LPA, min/day	186.3 (65.3)	193.1 (69.2)	6.8 (−5.6 to 19.2)	0.272
MPA, min/day	45.1 (27.2)	54.1 (28.3)	9.0 (2.8 to 15.1)	0.005
VPA, min/day	0.6 (1.7)	1.7 (3.5)	1.0 (0.01 to 2.1)	0.047
Steps, steps/day	6212 (2860)	7384 (2985)	1172 (365 to 1979)	0.006
ST, min/day	620.8 (92.0)	607.1 (94.9)	−13.7 (−34.0 to 6.6)	0.180
Valid day	4.4 (1.1)	4.8 (1.2)	0.4 (−0.1 to 0.9)	0.081
Physical activity outcomes on non-working day				
MVPA, min/day	63.7 (41.5)	77.4 (51.0)	13.7 (−2.0 to 29.5)	0.086
LPA, min/day	252.0 (90.0)	249.4 (93.9)	−2.6 (−33.8 to 28.6)	0.866
MPA, min/day	60.8 (41.4)	74.4 (51.7)	13.6 (−2.0 to 29.2)	0.086
VPA, min/day	3.0 (9.0)	3.1 (12.1)	0.1 (−1.5 to 1.7)	0.885
Steps, steps/day	8244 (4461)	9553 (4451)	1310 (63 to 2557)	0.040
ST, min/day	466.1 (108.4)	461.3 (121.2)	−4.8 (−49.2 to 39.6)	0.828
Valid day	2.9 (0.9)	2.2 (0.8)	−0.7 (−1.0 to −0.4)	<0.001

Notes: Data are presented as mean (standard deviation) for continuous variables or number (%) for categorical variables at weeks 0 and 8, and mean (95% CI) for change. Abbreviations: BMI, body mass index; CI, confidence interval; LPA, light physical activity; MPA, moderate-to-vigorous physical activity; MVPA, moderate-to-vigorous physical activity; ST, sedentary time; VPA, vigorous physical activity. * Missing data were mostly due to personal reasons (they did not want to answer).

**Table 4 ijerph-19-16859-t004:** Sub-domain analyses on commuting day and remote working day (*n* = 34).

	Week 0	Week 8	Change (95% CI)	*p*-Value
Characteristic				
Age, years	49.6 (9.4)			
Women, *n* (%)	8 (24)			
BMI, kg/m^2^; *n* = 31 *	23.0 (3.0)			
Physical activity outcomes on commuting working day				
MVPA, min/day	58.5 (27.5)	62.0 (25.3)	3.5 (−2.8 to 9.9)	0.263
LPA, min/day	179.7 (72.8)	189.3 (80.4)	9.6 (−9.2 to 28.4)	0.307
MPA, min/day	58.1 (27.6)	61.4 (25.0)	3.3 (−3.0 to 9.5)	0.292
VPA, min/day	0.4 (0.8)	0.6 (1.0)	0.2 (−0.02 to 0.5)	0.068
Steps, steps/day	8215 (2681)	8755 (2674)	540 (−274 to 1355)	0.186
ST, min/day	627.8 (98.8)	618.4 (87.8)	−9.4 (−32.9 to 14.0)	0.420
Valid day	2.1 (1.0)	2.4 (1.0)	0.3 (−0.1 to 0.8)	0.133
Physical activity outcomes on remote working day				
MVPA, min/day	34.1 (33.9)	41.5 (35.9)	7.1 (0.4 to 13.7)	0.037
LPA, min/day	175.9 (68.6)	170.7 (67.1)	−5.2 (−19.4 to 9.0)	0.460
MPA, min/day	32.9 (33.7)	40.2 (35.6)	7.3 (0.8 to 13.7)	0.028
VPA, min/day	1.5 (5.5)	1.3 (3.4)	−0.2 (−1.3 to 1.0)	0.758
Steps, steps/day	4261 (3526)	5087 (3498)	826 (46 to 1606)	0.039
ST, min/day	610.0 (104.6)	612.7 (95.1)	2.7 (−21.8 to 27.2)	0.823
Valid day	2.4 (1.3)	2.6 (1.2)	0.2 (−0.3 to 0.7)	0.439

Notes: Data are presented as mean (standard deviation) for continuous variables or number (%) for categorical variables at weeks 0 and 8, and mean (95% CI) for change. Abbreviations: BMI, body mass index; CI, confidence interval; LPA, light physical activity; MPA, moderate-to-vigorous physical activity; MVPA, moderate-to-vigorous physical activity; ST, sedentary time; VPA, vigorous physical activity. * Missing data were mostly due to personal reasons (they did not want to answer).

## Data Availability

The data presented in this study are available upon request from the corresponding author. The data were not publicly available because of privacy concerns.

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
