# Peer review of "Multi-Component Intervention to Promote Physical Activity in Japanese Office Workers: A Single-Arm Feasibility Study"

_ijerph, 2022, doi:10.3390/ijerph192416859_

Round 1

Reviewer 1 Report

The authors described the implementation procedure and the results of multi-component intervention to promote physical activity (PA). The results are presented for the entire group of test persons. It would be interesting to analyze the difference between people who responded positively to the intervention by increasing their physical activity from those who did not increase or decrease their physical activity during the observation period (8 weeks). Such analysis would help understand the reasons for the lack of PA increase and improve the intervention program.

Author Response

Thank you for your constructive comments. It is pertinent for the development of our future research. As you mentioned, it is essential to explore the differences between those who have increased and decreased physical activity during the intervention period. We tried to analyze the data according to your comment, and also tried to explore which component of the intervention was effective among the participants in this study. However, we could not identify the answers clearly. Hence, we have added the following sentence as a limitation (lines 351–354).

Third, we could not identify which component of the intervention program was effective; it may be dependent on the participants’ individual characteristics. Further qualitative studies could clarify this issue.

Reviewer 2 Report

Dear authors,

First of all, thank you for the opportunity to review this article.  The present article is focused in the assessment of changes in physical activity levels after a multimodal intervention within the work environment.

In general, the article is very clearly elaborated. It is a well conducted study; the aim and object of the research is understandable. The results are clearly presented and the discussion is clear. 

In my humble opinion, this is a suitable work for the journal and especially for the special issue Active and Sitting Time at Work—Evidence for Optimizing Worker Health. However, I would like you to better clarify some procedures:

The individual strategy involved providing baseline data to the participants, one of them being some sit-to-stand measurements, using a thigh-worn accelerometer (lines 133-134). However, section 2.3.1. (Lines 160-169) does not provide any information regarding this. Which measurements were collected?

Additionally, you also mention that the participants wore an accelerometer on the waist for 7 days (lines 172-175). This seems confusing to me: Did they always wear two accelerometers, or the thigh-worn accelerometer was only used on one occasion? And if this is the case, shouldn’t table 1 include that data? Please clarify.

 Also, it would be interesting to include data from the sit-to-stand measurements at the end of the intervention, for comparison with the baseline values.

One drawback of this study lies in its single-armed design, which is clearly stated as a limitation by the authors. Additionally, the authors point that a longer follow-up period is necessary to assess the validity of this intervention, which I appreciate. I would perhaps invite the authors to emphasize a little more the importance of longer follow-up periods to assess these types of interventions.

Finally, I would like to congratulate the authors for the work. I hope my comments and remarks helped!
